# The associations of *IGF2, IGF2R and IGF2BP2* gene polymorphisms with gestational diabetes mellitus: A case-control study

Wei Li[1☯], Lu She[2☯], Muyu Zhang[1☯], Mei Yang[3], Wenpei Zheng[1], Hua He[1], Ping Wang[1], Qiong Dai[1]*, Zhengtao Gong[1]*

1 Maternal and Child Health Hospital of Hubei Province, Tongji Medical College, Huazhong University of Science and Technology, Wuhan, China, 2 Xianning Center for Disease Control and Prevention, Xianning, China, 3 School of Public Health, Wuhan University of Science and Technology, Wuhan, China

☯ WL, LS and MZ are contributed equally to this work
* daiqiong74@163.com (QD); 11828116@163.com (ZG)

## Abstract

**Data Availability Statement:** The data that support the findings of this study is openly available in

### Objective

To investigate the associations of *Insulin-like growth factor-II* (*IGF2*) gene, *Insulin-like growth factor-II receptor* (*IGF2R*) gene and *Insulin-like growth factor-II binding protein 2* (*IGF2BP2*) gene polymorphisms with the susceptibility to gestational diabetes mellitus (GDM) in Chinese population.

### Methods

A total of 1703 pregnant women (835 GDM and 868 Non-GDM) were recruited in this case-control study. All participants underwent prenatal 75 g oral glucose tolerance test (OGTT) examinations during 24–28 gestational weeks at the Maternal and Child Health Hospital of Hubei Province from January 15, 2018 to March 31, 2019. Genotyping of candidate SNPs (*IGF2* rs680, *IGF2R* rs416572, *IGF2BP2* rs4402960, rs1470579, rs1374910, rs11705701, rs6777038, rs16860234, rs7651090) was performed on Sequenom MassARRAY platform. Logistic regression analysis was conducted to investigate the associations between candidate SNPs and risk of GDM. In addition, multifactor dimensionality reduction (MDR) method was applied to explore the effects of gene-gene interactions on GDM risk.

### Results

There were significant distribution differences between GDM group and non-GDM group in age, pre-pregnancy BMI, education level and family history of diabetes ($P < 0.05$). After adjusted for age, pre-pregnancy BMI, education level and family history of diabetes, there were no significant associations of the candidate SNPs polymorphisms and GDM risk ($P > 0.05$). Furthermore, there were no gene-gene interactions on the GDM risk among the candidate SNPs ($P > 0.05$). However, the fasting blood glucose (FBG) levels of rs6777038 CT carriers were significantly lower than TT carriers (4.69±0.69 vs. 5.03±1.57 mmol/L, $P <$

figshare at: https://doi.org/10.6084/m9.figshare.
25286227.v1.

**Funding:** This work was supported by the Health
Commission of Hubei Province (WJ2018H0134,
WJ2018H0145). The funders had role in data
collection and analysis, decision to publish and
preparation of the manuscript.

**Competing interests:** The authors have declared
that no competing interests exist.

0.01), and the OGTT-2h levels of rs6777038 CC and CT genotype carriers were significantly lower than TT genotype carriers (8.10±1.91 and 8.08±1.87 vs. 8.99±2.90 mmol/L, $P < 0.01$).

## Conclusions

*IGF2* rs680, *IGF2R* rs416572, *IGF2BP2* rs4402960, rs1470579, rs11705701, rs6777038, rs16860234, rs7651090 polymorphisms were not significantly associated with GDM risk in Wuhan, China. Further lager multicenter researches are needed to confirm these results.

## 1. Introduction

Gestational diabetes mellitus (GDM) is characterized by glucose intolerance with onset or first recognized during pregnancy, which is a common complication of pregnancy [1]. The incidence of GDM is gradually increasing worldwide, and the pooled incidence in China is as high as 14.8% [2]. The adverse outcomes associated with GDM in pregnant women and their offspring are diverse, such as preeclampsia, neonatal hypoglycemia, macrosomia, type 2 diabetes mellitus (T2DM) and so on [1,3,4]. Therefore, it is crucial to identify potential risk factors of GDM for the health of women and children.

As a status of pre-diabetic, GDM is generally considered to be similar to T2DM in pathogenic mechanism, which is related to β-cell dysfunction and insulin resistance (IR) [5–9]. Insulin-like growth factor-II (IGF2) is a peptide with a similar structure to insulin that promotes β-cell proliferation and survival, and it has been shown to be involved in the identification of insulin that regulates growth and metabolism [10,11]. Insulin-like growth factor-II receptor (IGF2R), also known as mannose-6-phosphate receptor (M6P), is the scavenger receptor of IGF2 and it is important for limiting the bioavailability of IGF2 and regulating its glucose regulatory activity [12]. The insulin-like growth factor-II binding protein 2 (IGF2BP2/IMP2) is one of the mRNA binding protein family translated by IGF2 and is highly expressed in islets [13].

Previous studies have shown that *IGF2* [14], *IGF2R* [15], *IGF2BP2* [16,17] gene polymorphisms are related to T2DM, but limited researches have been conducted in GDM. And so far, only the role of *IGF2BP2* polymorphism in GDM has been involved in genome-wide association studies (GWAS) in different populations [18,19]. Wu et al. conducted a meta-analysis in 2016 which showed that *IGF2BP2* rs4402960 was significantly associated with increased GDM risk [20], but negative conclusions were drew in Tarnowski's study subsequently [21]. The associations of *IGF2*, *IGF2R* and *IGF2BP2* gene polymorphisms with GDM risk are still ambiguous in Chinese population. Therefore, the purpose of the case-control study was to explore the associations between *IGF2*, *IGF2R*, *IGF2BP2* polymorphisms and the risk of GDM in Wuhan, China.

## 2. Materials and methods

### 2.1 Study population

Pregnant women who underwent prenatal examination at the Obstetrics and Gynecology Clinic of Maternal and Child Health Hospital of Hubei Province from January 15, 2018 to March 31, 2019 were consecutively enrolled in our study. After fasting for 8–12 hours, all subjects were given a routine 75 g oral glucose tolerance test (OGTT) at 24 to 28 gestational weeks. GDM was diagnosed when one or more following plasma values equaled or exceeded:

fasting blood glucose (FBG) $\geq$ 5.1 mmol/L (92 mg/dl), 1-hour blood glucose (OGTT-1h) $\geq$ 10.0 mmol/L (180 mg/dl), and 2-hour blood glucose (OGTT-2h) $\geq$ 8.5 mmol/L (153 mg/dl) [22]. The non-diabetic pregnant women matched with gestational weeks were randomly selected as the non-GDM group. Exclusion criteria were: age < 18 years, ethnic minorities, pre-gestational diabetes, multiple pregnancies, pregnancies complicated with endocrine diseases such as hypertension and polycystic ovary syndrome, any other medical condition that might affect glucose regulation, unable or unwilling to participate in the study, and samples failure genotyped. Finally, 1703 pregnant women (835 GDM and 868 Non-GDM) were recruited in this case-control study. All subjects were unrelated and lived in Wuhan, Hubei Province, central China.

The method of data collection was reported in a previous article [23]. According to the Chinese standard for obesity, pregnant women could be diagnosed as underweight (< 18.5 kg/m$^2$), normal (18.5 kg/m$^2$ $\leq$ BMI < 24 kg/m$^2$), overweight (24 kg/m$^2$ $\leq$ BMI < 28 kg/m$^2$) and obese ($\geq$ 28 kg/m$^2$). This study involving participants was reviewed and approved by the institutional review board of Wuhan University of Science and Technology, and was based on the principles of the Declaration of Helsinki. All participants provided their written informed consent for participation.

## 2.2 Selection and genotyping of SNPs

According to the results of T2DM GWAS, minor allele frequency (MAF) > 0.05 reported in the Chinese population, and tracking the latest results of *IGF2/IGF2R/IGFBP2* polymorphisms with GDM risk [20,24,25], we finally selected 9 SNPs (rs680, rs416572, rs4402960, rs1470579, rs1374910, rs11705701, rs6777038, rs16860234, rs7651090) that might be associated with the risk of GDM. Plasma glucose measurements were performed by glucose oxidase method on the Cobas 8000 Modular Analyzer Series (Roche, Mannheim). At recruitment, 2mL fasting peripheral venous blood was collected and placed in EDTA anticoagulant tube. After separation, it was packed in 1.5mL EP tube and stored at -80 ˚C until analysis. Genomic DNA was isolated from 0.5mL blood cells using approved guideline of the Relax Gene Blood DNA System DP348 (Tiangen, China). According to the manufacturer's instructions, the Sequenom MassARRAY platform (Sequenom, San Diego, California, USA) was used to genotype candidate SNPs, and a preliminary experiment was performed before formal genotyping. For quality control, 5% of duplicate samples were selected in a blind analysis. The call rates of rs680, rs416572, rs4402960, rs1470579, rs1374910, rs11705701, rs6777038, rs16860234, rs7651090 were 99.35%, 99.35%, 99.00%, 98.47%, 99.12%, 98.65%, 98.06%, 98.59% and 98.83% respectively, which were higher than the quality control standard (95%). In addition, the statistical power in this study was higher than 0.90.

## 2.3 Statistical analyses

The Kolmogorov-Smirnov test was used to test the normality of the distribution of continuous variables. Normal distribution data was expressed as mean ± standard deviation (SD), and differences among groups were compared by unpaired Student's *t*-test. Chi-square test was used for qualitative data. The Hardy-Weinberg Equilibrium test (HWE) was applied to test whether the participants were representative of the population. Chi-square test of goodness of fit was used to measure the coincidence between the observed number of genotypes and the HWE of all genotype frequencies at the locus. If *P* was above 0.05, the sample of this genotype conformed to the law of genetic equilibrium, which suggested that the sample had good population representation. SNPS that did not meet the HWE test were not included in subsequent analyses. The Box-Tidwell method was used to check the assumptions for the logistic

regression as age was a continuous variable, and the result showed there was a linear relationship between age and logit ($P$) ($P = 0.18 > 0.05$). Logistic regression was performed to evaluate the associations of genotypes and GDM risk. SHesis online software was used to analyze the linkage disequilibrium (LD) among *IGFBP2* gene SNPs and construct haplotypes (http://analysis.bio-x.cn/myAnalysis.php). The Multifactor dimension reduction (MDR) method was applied to evaluate gene-gene interactions on GDM risk. One-way ANOVA analysis was used to investigate the relationships between SNPs and blood glucose levels, and post hoc analyses were performed by the Least Significant Difference (LSD) method. G-Power 3.1 software was used for power analysis for the study. All the statistical analyses were conducted by SPSS Software, Version 26.0 (SPSS Inc., Chicago, IL, USA). $P < 0.05$ was accepted as statistically significant.

## 3. Results

### 3.1 Demographic and clinical characteristics

The demographic and clinical characteristics of the participants were shown in Table 1. Compared with non-GDM group, women with GDM had a higher FBG, OGTT-1h, OGTT-2h, age, pre-gestational BMI and education level ($P < 0.01$). In addition, the proportion of GDM patients with a family history of diabetes was higher than non-GDM patients (29.96% vs. 12.06%, $P < 0.01$).

**Table 1. Demographic and clinical characteristics of the study population.**

| Variables | GDM (n = 835) | Non-GDM (n = 868) | $\chi^2/t$ | $P$ |
|---|---|---|---|---|
| FBG (mmol/L) | 5.05±0.88 | 4.34±0.31 | 20.94 | **<0.01** |
| OGTT-1h (mmol/L) | 10.42±1.69 | 7.37±1.34 | 34.02 | **<0.01** |
| OGTT-2h (mmol/L) | 9.14±1.73 | 6.50±0.98 | 35.13 | **<0.01** |
| Age (year) | 30.97±4.56 | 28.84±4.21 | 9.99 | **<0.01** |
| Pre-pregnancy BMI (kg/m$^2$) | | | | |
| underweight (<18.5) | 100(12.55) | 134(21.27) | 54.66 | **<0.01** |
| normal (18.5≤BMI<24) | 492(61.73) | 423(67.14) | | |
| overweight (24≤BMI<28) | 152(19.07) | 58(9.21) | | |
| obese (≥28) | 53(6.65) | 15(2.38) | | |
| Education (years) | | | 33.86 | **<0.01** |
| ≤9 | 136(16.32) | 93(10.80) | | |
| 10–12 | 211(25.33) | 325(37.75) | | |
| ≥12 | 486(58.34) | 443(51.45) | | |
| Gravidity n (%) | | | 5.18 | 0.08 |
| 1 | 294(36.16) | 339(39.74) | | |
| 2 | 235(28.91) | 260(30.48) | | |
| ≥3 | 284(34.93) | 254(29.78) | | |
| Parity n (%) | | | 0.76 | 0.38 |
| Nulliparae (%) | 491(58.94) | 529(60.01) | | |
| Multiparae (%) | 342(41.06) | 338(38.99) | | |
| Family history of diabetes n (%) | | | 81.68 | **<0.01** |
| No | 582(70.04) | 751(87.94) | | |
| Yes | 249(29.96) | 103(12.06) | | |

Notes: FBG, fasting blood glucose; OGTT-1h, 1-hour blood glucose; OGTT-2h, 2-hour blood glucose; BMI, body mass index. Bold represents significance $P$ values.

### 3.2 Associations between candidate SNPs polymorphisms and GDM

The distributions of rs680, rs416572, rs4402960, rs1470579, rs11705701, rs6777038, rs16860234 and rs7651090 in the non-GDM group were all in HWE ($P > 0.05$), except for rs1374910 ($P < 0.05$). The genotype distributions of the remaining 8 candidate SNPs in the GDM and non-GDM groups and their associations with GDM were presented in Table 2. After adjusted for age, pre-gestational BMI, education level and family history of diabetes, there were no significant associations of the candidate SNPs polymorphisms and GDM risk ($P > 0.05$).

### 3.3 LD analysis and haplotype construction among SNPs of *IGFBP2* gene

The LD analysis among candidate SNPs of *IGF2BP2* gene was shown in Fig 1. The results showed that there was a strong LD among rs4402960, rs1470579 and rs7651090 ($D'>0.900$, $r^2>0.850$). Then haplotype construction was carried out, and the results showed that there were three haplotypes: GAA, TAG and TCG (Table 3). But no significant correlation was found between them and the risk of GDM.

### 3.4 Gene-gene interactions to GDM

Gene-gene interactions analysis indicated that both two-factor model (rs680, rs416572) and three-factor model (rs680, rs416572 and rs16860234) had good cross-validation consistency at 7/10, and the test accuracy of the two-factor model (0.50) was higher than the three-factor model (0.49), so the best model was the two-factor model. After further analysis, there was no significance of the test set in the two-factor gene-gene interactions ($P > 0.05$), as shown in Table 4.

### 3.5 Relationships between candidate SNPs polymorphisms and glycemic levels

Table 5 presented that the FBG levels of rs6777038 CT carriers were significantly lower than TT genotypes carriers (4.69±0.69 vs. 5.03±1.57 mmol/L, $P < 0.01$), and the OGTT-2h levels of rs6777038 CC and CT genotype carriers were significantly lower than TT genotype carriers (8.10±1.91 and 8.08±1.87 vs. 8.99±2.90 mmol/L, $P < 0.01$).

## 4. Discussion

In this case-control study, we analyzed the associations of eight candidate SNPs polymorphisms (*IGF2* rs680, *IGF2R* rs416572, *IGF2BP2* rs4402960, rs1470579, rs11705701, rs6777038, rs16860234, rs7651090) with GDM risk, except for rs1374910, which was excluded due to not meet with the HWE test. The results showed that candidate SNPs polymorphisms were not associated with increased GDM risk in Wuhan, China. Moreover, there were no gene-gene interactions on the GDM risk among the candidate SNPs.

*IGF2* gene located in chromosome 11p15.5 region, the same as insulin [26]. *IGF2* can promote pancreatic β-cell growth and re-expression during cell replication, renewal and apoptosis [11]. Although *IGF2* is an important insulin signaling molecule, there are few studies on the correlations between rs680 polymorphism and T2DM [24,27,28]. Khan et al. showed that rs680 was associated with T2DM risk [27,28], but had no significant correlations with GDM in India [11], which was consistent with our results in Chinese population.

As the scavenger of *IGF2*, *IGF2R* has been shown to play a key role in glucose metabolism [29,30]. *IGF2R* is located on chromosome 6q26, a region that contains a genetic marker associated with insulin resistance traits in Mexican—Americans [30]. A previous study has shown

**Table 2. Associations between candidate SNPs polymorphisms and GDM.**

| Gene | | Genotypes | GDM n (%) | Non-GDM n (%) | Unadjusted OR (95%CI) | Adjusted OR (95%CI) * |
|---|---|---|---|---|---|---|
| *IGF2* | | | | | | |
| rs680 | codominant model | TT | 237(28.59) | 271(31.40) | 1.00(ref.) | 1.00(ref.) |
| | | TC | 415(50.06) | 405(46.93) | 1.17(0.94–1.46) | 1.18(0.91–1.53) |
| | | CC | 177(21.35) | 187(21.67) | 1.08(0.83–1.42) | 1.15(0.84–1.58) |
| | dominant model | TC+CC | 692(71.41) | 592(68.60) | 1.14(0.93–1.41) | 1.17(0.92–1.49) |
| | | TT | 237(28.59) | 271(31.40) | 1.00(ref.) | 1.00(ref.) |
| | recessive model | CC | 177(21.35) | 187(21.67) | 0.98(0.78–1.24) | 1.04(0.79–1.37) |
| | | TT+TC | 652(78.65) | 676(78.33) | 1.00(ref.) | 1.00(ref.) |
| *IGF2R* | | | | | | |
| rs416572 | codominant model | CC | 571(68.80) | 605(70.19) | 1.00(ref.) | 1.00(ref.) |
| | | CT | 237(28.55) | 231(26.80) | 1.09(0.88–1.35) | 1.07(0.83–1.38) |
| | | TT | 22(2.65) | 26(3.02) | 0.90(0.50–1.60) | 0.75(0.39–1.45) |
| | dominant model | CT+TT | 259(31.20) | 27(29.81) | 1.07(0.87–1.31) | 1.03(0.81–1.32) |
| | | CC | 571(68.80) | 605(70.19) | 1.00(ref.) | 1.00(ref.) |
| | recessive model | TT | 22(2.65) | 26(3.02) | 0.88(0.49–1.56) | 0.74(0.39–1.42) |
| | | CC+CT | 809(97.35) | 836(96.98) | 1.00(ref.) | 1.00(ref.) |
| *IGF2BP2* | | | | | | |
| rs4402960 | codominant model | GG | 445(54.00) | 488(56.61) | 1.00(ref.) | 1.00(ref.) |
| | | GT | 321(38.96) | 318(36.89) | 1.11(0.91–1.35) | 1.17(0.92–1.48) |
| | | TT | 58(7.04) | 56(6.50) | 1.14(0.77–1.68) | 1.10(0.68–1.75) |
| | dominant model | GT+TT | 379(46.00) | 374(43.39) | 1.11(0.92–1.35) | 1.16(0.92–1.45) |
| | | GG | 445(54.00) | 488(56.61) | 1.00(ref.) | 1.00(ref.) |
| | recessive model | TT | 58(7.04) | 56(6.50) | 1.09(0.75–1.59) | 1.03(0.65–1.63) |
| | | GG+GT | 766(92.96) | 806(93.50) | 1.00(ref.) | 1.00(ref.) |
| rs1470579 | codominant model | AA | 430(52.50) | 473(55.13) | 1.00(ref.) | 1.00(ref.) |
| | | AC | 323(39.44) | 321(37.41) | 1.11(0.90–1.36) | 1.19(0.94–1.51) |
| | | CC | 66(8.06) | 64(7.46) | 1.13(0.79–1.64) | 1.05(0.68–1.64) |
| | dominant model | AC+CC | 389(47.50) | 385(44.87) | 1.11(0.92–1.35) | 1.17(0.64–1.51) |
| | | AA | 430(52.50) | 473(55.13) | 1.00(ref.) | 1.00(ref.) |
| | recessive model | CC | 66(8.06) | 64(7.46) | 1.09(0.76–1.56) | 0.98(0.64–1.51) |
| | | AA+AC | 753(91.94) | 794(92.54) | 1.00(ref.) | 1.00(ref.) |
| rs11705701 | codominant model | GG | 465(56.64) | 512(59.60) | 1.00(ref.) | 1.00(ref.) |
| | | GA | 305(37.15) | 301(35.04) | 1.12(0.91–1.37) | 1.21(0.95–1.54) |
| | | AA | 51(6.21) | 46(5.36) | 1.22(0.80–1.85) | 1.02(0.62–1.68) |
| | dominant model | GA+AA | 356(43.36) | 347(40.40) | 1.13(0.93–1.37) | 1.18(0.94–1.49) |
| | | GG | 465(56.64) | 512(59.60) | 1.00(ref.) | 1.00(ref.) |
| | recessive model | AA | 51(6.21) | 46(5.36) | 1.71(0.78–1.77) | 0.95(0.59–1.55) |
| | | GG+GA | 770(93.79) | 813(94.64) | 1.00(ref.) | 1.00(ref.) |
| rs6777038 | codominant model | CC | 547(67.20) | 571(66.71) | 1.00(ref.) | 1.00(ref.) |
| | | CT | 230(28.26) | 258(30.14) | 0.93(0.75–1.15) | 1.00(0.77–1.28) |
| | | TT | 37(4.55) | 27(3.15) | 1.43(0.86–2.38) | 1.32(0.72–2.42) |
| | dominant model | CT+TT | 267(32.80) | 285(33.29) | 0.98(0.80–1.20) | 1.03(0.81–1.31) |
| | | CC | 547(67.20) | 571(66.71) | 1.00(ref.) | 1.00(ref.) |
| | recessive model | TT | 37(4.55) | 27(3.15) | 1.46(0.88–2.42) | 1.32(0.72–2.41) |
| | | CC+CT | 777(95.45) | 829(96.85) | 1.00(ref.) | 1.00(ref.) |

(*Continued*)

**Table 2.** (Continued)

| Gene | | Genotypes | GDM n (%) | Non-GDM n (%) | Unadjusted OR (95%CI) | Adjusted OR (95%CI) * |
|---|---|---|---|---|---|---|
| rs16860234 | codominant model | AA | 522(63.74) | 573(66.63) | 1.00(ref.) | 1.00(ref.) |
| | | AC | 267(32.60) | 254(29.53) | 1.15(0.94–1.42) | 1.19(0.93–1.52) |
| | | CC | 30(3.66) | 33(3.84) | 0.99(0.60–1.66) | 0.98(0.52–1.85) |
| | dominant model | AC+CC | 297(36.26) | 287(33.37) | 1.14(0.93–1.39) | 1.17(0.92–1.48) |
| | | AA | 522(63.74) | 573(66.63) | 1.00(ref.) | 1.00(ref.) |
| | recessive model | CC | 30(3.66) | 33(3.84) | 0.95(0.58–1.58) | 0.93(0.49–1.74) |
| | | AA+AC | 789(96.34) | 827(96.16) | 1.00(ref.) | 1.00(ref.) |
| rs7651090 | codominant model | AA | 445(54.00) | 486(56.58) | 1.00(ref.) | 1.00(ref.) |
| | | AG | 321(38.96) | 317(36.90) | 1.11(0.90–1.35) | 1.17(0.92–1.48) |
| | | GG | 58(7.04) | 56(6.52) | 1.13(0.77–1.67) | 1.09(0.68–1.74) |
| | dominant model | AG+GG | 379(46.00) | 373(43.42) | 1.11(0.92–1.35) | 1.16(0.92–1.45) |
| | | AA | 445(54.00) | 486(56.58) | 1.00(ref.) | 1.00(ref.) |
| | recessive model | GG | 58(7.04) | 56(6.52) | 1.09(0.74–1.59) | 1.02(0.65–1.62) |
| | | AA+AG | 766(92.96) | 803(93.48) | 1.00(ref.) | 1.00(ref.) |

Notes:

*Logistic regression analyses adjusted for age, pre-gestational BMI and family history of diabetes; Ref, reference genotype.

that rs416572 in *IGF2R* 3'UTR region may be associated with the risk of T2DM [15,30]. But our results showed that rs416572 was not associated with GDM risk in Wuhan, China.

*IGF2BP2* is located on chromosome 3q27 and binds to the 5'-UTR regions of IGF2 mRNA [31]. *IGF2BP2* regulates pancreatic β cell function by inhibiting the first stage of insulin secretion. Previous studies have shown that *IGF2BP2* rs4402960 [32], rs1470579 [33,34] and rs11705701 [33] were correlated with T2DM. But Xie et al. reported a negative result between rs1470579 and GDM risk in Chinese population [35], and Tarnowski et al. showed that rs11705701 was also not correlated with the risk of GDM in Poland [21], both of which were consistent with our research results. The associations between rs4402960 polymorphism and the risk of GDM remains controversial [20,21,36–38]. Previous studies have shown that rs4402960 is a susceptibility gene locus for GDM in Japan [36] and Korea [37], but this conclusion has not been obtained in Russia population [38]. Our results also showed that rs4402960 was not associated with the risk of GDM in Chinese population, which was consistent with the conclusion of recent meta-analysis [38,39].

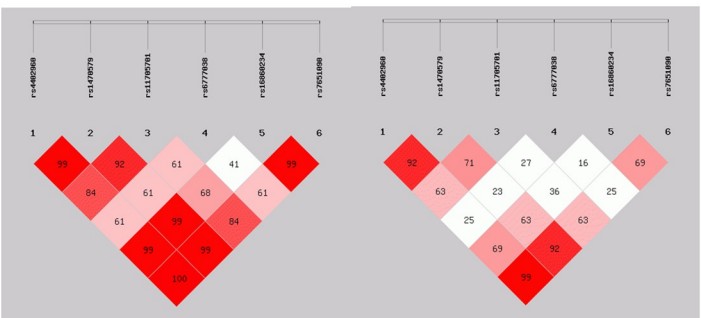

**Fig 1. Inter-single nucleotide polymorphism linkage disequilibrium analysis of IGF2BBP2 gene (*D'* test on the left, *r²* test on the right).**

**Table 3. Haplotype analysis for rs4402960, rs1470579 and rs7651090 of *IGF2BP2* gene.**

| Haplotype | GDM *n*(%) | Non-GDM *n*(%) | $\chi^2$ | *P* | OR(95%CI) |
|---|---|---|---|---|---|
| GAA | 430.00(26.35) | 422.99(24.79) | 1.059 | 0.303 | 1.085(0.929–1.268) |
| TAG | 25.00(1.53) | 26.01(1.52) | <0.001 | 0.987 | 1.005(0.578–1.747) |
| TCG | 1177.00(72.12) | 1255.98(73.62) | 0.952 | 0.329 | 0.927(0.796–1.080) |

**Table 4. Interactions of two-factor gene-gene model.**

|  | Sensitivity | Specificity | $\chi^2$ | *P* | OR (95%CI) | Kappa |
|---|---|---|---|---|---|---|
| Training set | 0.45 | 0.63 | 9.86 | <0.01 | 1.39 (1.13,1.70) | 0.08 |
| Test set | 0.40 | 0.60 | 0.01 | **0.94** | 1.02 (0.55,1.89) | 0.01 |
| Total set | 0.54 | 0.54 | 9.97 | <0.01 | 1.36 (1.12,1.65) | 0.08 |

**Table 5. Associations between candidate SNPs polymorphisms and glycemic levels.**

| Gene | SNPs | Variables | Genotypes | | | *F* | *P* |
|---|---|---|---|---|---|---|---|
|  |  |  | AA | AB | BB |  |  |
| *IGF2* | rs680 | FBG | 4.78±0.79 | 4.78±0.72 | 4.80±0.98 | 0.07 | 0.93 |
|  |  | OGTT-1h | 9.14±2.19 | 9.36±2.04 | 9.24±2.35 | 1.23 | 0.29 |
|  |  | OGTT-2h | 8.06±1.92 | 8.18±1.87 | 8.13±2.23 | 0.40 | 0.67 |
| *IGF2R* | rs416572 | FBG | 4.81±0.84 | 4.76±0.72 | 4.58±0.59 | 1.76 | 0.17 |
|  |  | OGTT-1h | 9.28±2.20 | 9.25±2.06 | 9.25±1.86 | 0.02 | 0.98 |
|  |  | OGTT-2h | 8.16±1.98 | 8.12±1.98 | 7.73±1.42 | 0.84 | 0.43 |
| *IGF2BP2* | rs4402960 | FBG | 4.76±0.73 | 4.82±0.86 | 4.81±1.00 | 0.87 | 0.42 |
|  |  | OGTT-1h | 9.21±2.13 | 9.33±2.16 | 9.44±2.38 | 0.76 | 0.47 |
|  |  | OGTT-2h | 8.08±1.90 | 8.18±2.03 | 8.33±2.22 | 0.81 | 0.45 |
|  | rs1470579 | FBG | 4.76±0.73 | 4.81±0.82 | 4.80±0.95 | 0.50 | 0.61 |
|  |  | OGTT-1h | 9.20±2.13 | 9.35±2.17 | 9.41±2.32 | 0.91 | 0.40 |
|  |  | OGTT-2h | 8.07±1.92 | 8.20±2.01 | 8.28±2.18 | 0.86 | 0.42 |
|  | rs11705701 | FBG | 4.77±0.75 | 4.80±0.85 | 4.83±1.00 | 0.24 | 0.78 |
|  |  | OGTT-1h | 9.22±2.14 | 9.34±2.15 | 9.24±2.35 | 0.44 | 0.64 |
|  |  | OGTT-2h | 8.09±1.96 | 8.20±2.00 | 8.19±2.01 | 0.40 | 0.67 |
|  | rs6777038 | FBG | 4.80±0.74 | 4.69±0.69* | 5.03±1.57 | 5.46 | **<0.01** |
|  |  | OGTT-1h | 9.25±2.10 | 9.17±2.09 | 9.84±3.11 | 2.25 | 0.11 |
|  |  | OGTT-2h | 8.10±1.91* | 8.08±1.87* | 8.99±2.90 | 5.46 | **<0.01** |
|  | rs16860234 | FBG | 4.76±0.71 | 4.82±0.91 | 4.96±1.18 | 1.86 | 0.16 |
|  |  | OGTT-1h | 9.19±2.10 | 9.40±2.20 | 9.44±2.70 | 1.41 | 0.24 |
|  |  | OGTT-2h | 8.07±1.88 | 8.23±2.08 | 8.44±2.56 | 1.54 | 0.22 |
|  | rs7651090 | FBG | 4.76±0.73 | 4.82±0.86 | 4.81±1.00 | 0.77 | 0.46 |
|  |  | OGTT-1h | 9.22±2.12 | 9.34±2.15 | 9.44±2.38 | 0.69 | 0.50 |
|  |  | OGTT-2h | 8.10±1.90 | 8.18±2.03 | 8.33±2.22 | 0.67 | 0.51 |

Notes: AA: Homozygote of major allele; AB: Heterozygote; BB: Homozygote of minor allele.

* Compared with BB Genotype by LSD test, $P < \alpha' = 0.05/3$.

To our knowledge, limited studies have been conducted to explore the correlations of rs6777038, rs16860234 and rs7651090 polymorphisms with GDM [40]. Only Salem showed that these three loci were associated with glutamic acid decarboxylase antibodies (GADA) negative diabetes [40]. The results of this study showed that these three candidate loci were not associated with the risk of GDM. In addition, the results of this study also showed that there were significant differences in FBG and OGTT-2h levels of rs6777038 genotypes, indicating that blood glucose only had a partial effect on the occurrence of GDM.

As disease susceptibility can't be attributed to a single polymorphism or allele, but rather to a combination of multiple polymorphisms [41–44]. We evaluated gene-gene interactions using multifactor dimension reduction (MDR), a novel method to examine the combined effects of multiple factors in disease susceptibility [45]. The results showed that there were no gene-gene interactions on the GDM risk among the candidate SNPs, which needs to be further verified.

Our study has the following strengths. First, we selected 8 SNPs to explore their correlations with GDM, and as far as we know this is the first study to detect the relationships between *IGF2R* rs416572, *IGF2BP2* rs7651090, rs6777038, rs16860234 and GDM. Second, we used MDR method to explore the gene-gene interactions on the GDM risk among the candidate SNPs. Finally, the sample size of this study is relatively large and the statistical power for each SNP is higher than 0.90, so the results are reliable.

However, there are some limitations in this study. First, the subjects were all from hospitals, which may exist admission rate bias, and further population-based studies will be needed. Second, we did not eliminate the effects of all confounding factors, such as dietary habits and exercise patterns. Third, since our results were all negative, clinicians still could not identify high-risk individuals by these sites. Finally, the levels and activities of *IGF2, IGF2R* and *IGF2BP2* were not detected in this study. Therefore, more studies will be needed in the future to verify the results of this study.

## 5. Conclusions

In conclusion, the study revealed that *IGF2* rs680, *IGF2R* rs416572, *IGF2BP2* rs4402960, rs1470579, rs11705701, rs6777038, rs16860234, rs7651090 polymorphisms were not significantly associated with GDM in Wuhan, China. Further lager multicenter researches are needed to confirm these results.

## Author Contributions

**Conceptualization:** Mei Yang, Zhengtao Gong.

**Data curation:** Wei Li, Lu She, Muyu Zhang, Wenpei Zheng, Hua He, Ping Wang.

**Investigation:** Mei Yang, Wenpei Zheng, Hua He, Ping Wang.

**Methodology:** Wei Li, Muyu Zhang.

**Project administration:** Qiong Dai.

**Software:** Lu She, Muyu Zhang.

**Supervision:** Qiong Dai, Zhengtao Gong.

**Validation:** Mei Yang.

**Writing – original draft:** Wei Li, Lu She.

**Writing – review & editing:** Qiong Dai, Zhengtao Gong.

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
