## [Decision Letter · Decision Letter 0]

7 Nov 2023

PONE-D-23-27799The associations of IGF2, IGF2R and IGF2BP2 gene polymorphisms with gestational diabetes mellitus: a case-control studyPLOS ONE

Dear Dr. Dai,

Thank you for submitting your manuscript to PLOS ONE. After careful consideration, we feel that it has merit but does not fully meet PLOS ONE’s publication criteria as it currently stands. Therefore, we invite you to submit a revised version of the manuscript that addresses the points raised during the review process.

In this study the authors tested associations between SNPs in IGF2-related genes and GDM in pregnant women from China. It is reasonably well written, but there are a number of revisions required in the manuscript, as highlighted by the reviewers and by the comments below. To be publishable in this journal publication criteria have to be followed and the writing has to be consistent with STROBE guidelines for genetic association studies. POINTS REQUIRING REVISION (in addition to those from the reviewers):1. One of the publication criteria for this journal is that the data is available without restriction. Therefore, I would suggest upoloading the anonymised data to a repository and including the hyperlink to that in the manuscript (it is not sufficient to say that the data is available from the corresponding author).2. The main issue that I have with the results are that they were tested adjusted for BMI and family history of diabetes, both factors which are generally associated with GDM. Therefore if a SNP was associated with GDM but the association was mediated through BMI (or family history), by adjusting for it, the association may have been weakened. This could explain some of the 'negative' results. I would therefore suggest presenting results both unadjusted, and adjusted for potential confounders.3. Lines 34, 73, 115. Please change the word "correlations" to "associations."4. How were the blood glucose concentrations measured in the OGTT?5. For how long were the women who underwent OGTTs fasted?6. In the hospital where this study took place, do all pregnant women undergo OGTTs or is it just women with risk factors for GDM? This is important to state, as it could introduce biases into the recruitment of controls if it were women with GDM risk factors.7. What biological sample (blood/saliva/swab) was taken for the extraction of DNA? When was the sample taken? How was the DNA extracted?8. Line 98. Please change the phrase "track the literatures" to something more meaningful.9. Line 100. Was the genotyping performed according to the manufacturer's instructions?10. Line 105. It is the statistical test that contains the statistical power, not the SNP itself.11. Line 118. Please change "multiple comparisons" to "post hoc analyses."12. How was missing data dealt with?13. Was there any significant linkage disequilibrium between the different IGF2BP2 SNPs? If so, how was this dealt with?14. Line 203. Please change "non-significant associations" to "not significantly associated..." ==============================

We look forward to receiving your revised manuscript.

Kind regards,

Clive J. Petry, PhD

Academic Editor

PLOS ONE

Journal Requirements:

"This work was supported by the Health Commission of Hubei Province (WJ2018H0134, WJ2018H0145)."

Reviewers' comments:

Reviewer's Responses to Questions

**Comments to the Author**

1. Is the manuscript technically sound, and do the data support the conclusions?

Reviewer #1: Yes

Reviewer #2: Yes

2. Has the statistical analysis been performed appropriately and rigorously? 

Reviewer #1: No

Reviewer #2: Yes

3. Have the authors made all data underlying the findings in their manuscript fully available?

Reviewer #1: Yes

Reviewer #2: No

4. Is the manuscript presented in an intelligible fashion and written in standard English?

Reviewer #1: Yes

Reviewer #2: Yes

5. Review Comments to the Author

Reviewer #1: Li et al. investigated the association of IGF2, IGF2R, and IGF2BP2 gene polymorphisms with gestational diabetes mellitus (GDM). The manuscript is well-written and well-analyzed and can be accepted following a major revision.

My comments are:

• In the abstract, the conclusion part should be rewritten and expanded.

• In the introduction, the authors should discuss more relevant publications in terms of the pathophysiology, incidence, risk factors, and SNPs already associated with diabetes by citing the following references one by one:

DOI: 10.1038/s41598-023-33239-3 - 10.1007/s40995-021-01229-7 - 10.1007/s40200-021-00740-3 - 10.1007/s13410-020-00898-1 - 10.1016/j.genrep.2018.06.019 - 10.1007/s40200-020-00590-5 -10.1155/2022/4327342

• The names of all genes must be italicized, and fonts must also be consistent throughout the text.

• In Table 2, Please also include the dominant and recessive inheritance models.

• The authors stated that "the distributions ofrs1374910 in the non-GDM group deviated from HWE. How do you justify that? Because such deviation essentially reflects improper selection of controls.

• How do these findings help clinicians to manage GDM? It should be discussed along with the limitations of the study.

• The conclusion should be expanded and include a "Take to home" message.

I would like to review the revised version of the manuscript

.

Reviewer #2: In this study, the authors found no association of IGF2, IGF2R, and IGF2BP2 polymorphisms with gestational diabetes mellitus in Wuhan, China. The study is very well designed and written.

1. More details on the genotyping method are required.

2. The language of the manuscript is acceptable, however, further revision might be required.

6. PLOS authors have the option to publish the peer review history of their article (what does this mean?). If published, this will include your full peer review and any attached files.

Reviewer #1: No

Reviewer #2: No

---

## [Author Response · Author response to Decision Letter 0]

17 Dec 2023

POINTS REQUIRING REVISION (in addition to those from the reviewers):

1. One of the publication criteria for this journal is that the data is available without restriction. Therefore, I would suggest upoloading the anonymised data to a repository and including the hyperlink to that in the manuscript (it is not sufficient to say that the data is available from the corresponding author).

Response: Thank the editor very much for this advice. The anonymized data was unloaded: https://pan.baidu.com/s/1TEpjmm8t-WXK-tS8Qr4KVw?pwd=1234 (Line 245)

2. The main issue that I have with the results are that they were tested adjusted for BMI and family history of diabetes, both factors which are generally associated with GDM. Therefore, if a SNP was associated with GDM but the association was mediated through BMI (or family history), by adjusting for it, the association may have been weakened. This could explain some of the 'negative' results. I would therefore suggest presenting results both unadjusted, and adjusted for potential confounders.

Response: Thank the editor very much for this advice. The unadjusted results were presented in Table 2, and there were no significant associations of the candidate SNPs polymorphisms and GDM risk (P > 0.05). 

3. Lines 34, 73, 115. Please change the word "correlations" to "associations."

Response: Thank the editor very much for this advice. The imprecise statement has been modified in the revised manuscript. (Line 35, 77, 129)

4. How were the blood glucose concentrations measured in the OGTT?

Response: Thank the editor very much for this advice. Plasma glucose measurements were performed by glucose oxidase method on the Cobas 8000 Modular Analyzer Series (Roche, Mannheim). (Line 105)

5. For how long were the women who underwent OGTTs fasted?

Response: Thank the editor very much for this advice. The pregnant women who underwent OGTTs should fast for 8-12 hours. (Line 83)

6. In the hospital where this study took place, do all pregnant women undergo OGTTs or is it just women with risk factors for GDM? This is important to state, as it could introduce biases into the recruitment of controls if it were women with GDM risk factors.

Response: Thank the editor very much for this advice. In China, OGTT is a routine test during 24 to 28 gestational weeks. All pregnant women undergo OGTTs in the Obstetrics and Gynecology Clinic. (Line 83)

7. What biological sample (blood/saliva/swab) was taken for the extraction of DNA? When was the sample taken? How was the DNA extracted?

Response: Thank the editor very much for this advice. At recruitment, 2mL fasting peripheral venous blood was collected and placed in EDTA anticoagulant tube. After separation, it was packed in 1.5mL EP tube and stored at -80℃ until analysis. Genomic DNA was isolated from 0.5mL blood cells using approved guideline of the Relax Gene Blood DNA System DP348 (Tiangen, China). (Line 106)

8. Line 98. Please change the phrase "track the literatures" to something more meaningful.

Response: Thank the editor very much for this advice. The imprecise statement has been modified in the revised manuscript. (Line 102)

9. Line 100. Was the genotyping performed according to the manufacturer's instructions?

Response: Thank the editor very much for this advice. The process of genotyping was performed according to the manufacturer’s instructions. (Line 109)

10. Line 105. It is the statistical test that contains the statistical power, not the SNP itself.

Response: Thank the editor very much for this advice. The imprecise statement has been modified in the revised manuscript. (Line 115)

11. Line 118. Please change "multiple comparisons" to "post hoc analyses."

Response: Thank the editor very much for this advice. This sentence has been revised. (Line 133)

12. How was missing data dealt with?

Response：Thank the editor for this suggestion. For genotyping data, if all the results of a sample fail to be evenly typed, the sample is deleted. If not, the results of the failed typing are included as blank values, the same with the demographic information missing. Since there are very few missing values in this study, blank value has little effect on the overall results. (Line 91)

13. Was there any significant linkage disequilibrium between the different IGF2BP2 SNPs? If so, how was this dealt with?

Response: Thank the editor very much for this advice. The LD analysis among candidate SNPs of IGF2BP2 gene was shown in Figure 1. The results showed that there was a strong LD among rs4402960, rs1470579 and rs7651090 (D’>0.900, r2>0.850). Then haplotype construction was carried out, and the results showed that there were three haplotypes: GAA, TAG and TCG (Table 3). But no significant correlation was found between them and the risk of GDM. (Line 129-131 and 151-156)

Figure 1 Inter-single nucleotide polymorphism linkage disequilibrium analysis of IGF2BBP2 gene (D’ test on the left, r2 test on the right)

Table 3. Haplotype analysis for rs4402960, rs1470579 and rs7651090 of IGF2BP2 gene 

Haplotype GDM n(%) Non-GDM n(%) χ2 P OR(95%CI)

GAA 430.00(26.35) 422.99(24.79) 1.059 0.303 1.085(0.929-1.268)

TAG 25.00(1.53) 26.01(1.52) <0.001 0.987 1.005(0.578-1.747)

TCG 1177.00(72.12) 1255.98(73.62) 0.952 0.329 0.927(0.796-1.080)

14. Line 203. Please change "non-significant associations" to "not significantly associated..."

Response：Thank the editor very much for this advice. This sentence has been revised. (Line 232)

Reviewer #1: Li et al. investigated the association of IGF2, IGF2R, and IGF2BP2 gene polymorphisms with gestational diabetes mellitus (GDM). The manuscript is well-written and well-analyzed and can be accepted following a major revision.

My comments are:

1. In the abstract, the conclusion part should be rewritten and expanded.

Response: Thank the reviewer very much for this advice. The conclusion part in the Abstract has been revised. （Line 47）

2. In the introduction, the authors should discuss more relevant publications in terms of the pathophysiology, incidence, risk factors, and SNPs already associated with diabetes by citing the following references one by one: DOI: 10.1038/s41598-023-33239-3 - 10.1007/s40995-021-01229-7 - 10.1007/s40200-021-00740-3 - 10.1007/s13410-020-00898-1 - 10.1016/j.genrep.2018.06.019 - 10.1007/s40200-020-00590-5 -10.1155/2022/4327342

Response: Thank the reviewer very much for this advice. In the revised manuscript, the introduction has been enriched and the above-mentioned references has been cited. 

3.The names of all genes must be italicized, and fonts must also be consistent throughout the text.

Response: Thank the reviewer very much for this advice. The format has been retouched in the revised manuscript. 

4. In Table 2, Please also include the dominant and recessive inheritance models.

Response: Thank the reviewer very much for this advice. The dominant and recessive inheritance models have been included in revised Table 2. 

Table 2. Associations between candidate SNPs polymorphisms and GDM.

Gene Genotypes GDM

n (%) Non-GDM

n (%) Unadjusted OR (95%CI) Adjusted OR (95%CI) *

IGF2 

rs680 codominant model TT 237(28.59) 271(31.40) 1.00(ref.) 1.00(ref.)

 TC 415(50.06) 405(46.93) 1.17(0.94-1.46) 1.18(0.91-1.53)

 CC 177(21.35) 187(21.67) 1.08(0.83-1.42) 1.15(0.84-1.58)

 dominant model TC+CC 692(71.41) 592(68.60) 1.14(0.93-1.41) 1.17(0.92-1.49)

 TT 237(28.59) 271(31.40) 1.00(ref.) 1.00(ref.)

 recessive model CC 177(21.35) 187(21.67) 0.98(0.78-1.24) 1.04(0.79-1.37)

 TT+TC 652(78.65) 676(78.33) 1.00(ref.) 1.00(ref.)

IGF2R 

rs416572 codominant model CC 571(68.80) 605(70.19) 1.00(ref.) 1.00(ref.)

 CT 237(28.55) 231(26.80) 1.09(0.88-1.35) 1.07(0.83-1.38)

 TT 22(2.65) 26(3.02) 0.90(0.50-1.60) 0.75(0.39-1.45)

 dominant model CT+TT 259(31.20) 27(29.81) 1.07(0.87-1.31) 1.03(0.81-1.32)

 CC 571(68.80) 605(70.19) 1.00(ref.) 1.00(ref.)

 recessive model TT 22(2.65) 26(3.02) 0.88(0.49-1.56) 0.74(0.39-1.42)

 CC+CT 809(97.35) 836(96.98) 1.00(ref.) 1.00(ref.)

IGF2BP2 

rs4402960 codominant model GG 445(54.00) 488(56.61) 1.00(ref.) 1.00(ref.)

 GT 321(38.96) 318(36.89) 1.11(0.91-1.35) 1.17(0.92-1.48)

 TT 58(7.04) 56(6.50) 1.14(0.77-1.68) 1.10(0.68-1.75)

 dominant model GT+TT 379(46.00) 374(43.39) 1.11(0.92-1.35) 1.16(0.92-1.45)

 GG 445(54.00) 488(56.61) 1.00(ref.) 1.00(ref.)

 recessive model TT 58(7.04) 56(6.50) 1.09(0.75-1.59) 1.03(0.65-1.63)

 GG+GT 766(92.96) 806(93.50) 1.00(ref.) 1.00(ref.)

rs1470579 codominant model AA 430(52.50) 473(55.13) 1.00(ref.) 1.00(ref.)

 AC 323(39.44) 321(37.41) 1.11(0.90-1.36) 1.19(0.94-1.51)

 CC 66(8.06) 64(7.46) 1.13(0.79-1.64) 1.05(0.68-1.64)

 dominant model AC+CC 389(47.50) 385(44.87) 1.11(0.92-1.35) 1.17(0.64-1.51)

 AA 430(52.50) 473(55.13) 1.00(ref.) 1.00(ref.)

 recessive model CC 66(8.06) 64(7.46) 1.09(0.76-1.56) 0.98(0.64-1.51)

 AA+AC 753(91.94) 794(92.54) 1.00(ref.) 1.00(ref.)

rs11705701 codominant model GG 465(56.64) 512(59.60) 1.00(ref.) 1.00(ref.)

 GA 305(37.15) 301(35.04) 1.12(0.91-1.37) 1.21(0.95-1.54)

 AA 51(6.21) 46(5.36) 1.22(0.80-1.85) 1.02(0.62-1.68)

 dominant model GA+AA 356(43.36) 347(40.40) 1.13(0.93-1.37) 1.18(0.94-1.49)

 GG 465(56.64) 512(59.60) 1.00(ref.) 1.00(ref.)

 recessive model AA 51(6.21) 46(5.36) 1.71(0.78-1.77) 0.95(0.59-1.55)

 GG+GA 770(93.79) 813(94.64) 1.00(ref.) 1.00(ref.)

rs6777038 codominant model CC 547(67.20) 571(66.71) 1.00(ref.) 1.00(ref.)

 CT 230(28.26) 258(30.14) 0.93(0.75-1.15) 1.00(0.77-1.28)

 TT 37(4.55) 27(3.15) 1.43(0.86-2.38) 1.32(0.72-2.42)

 dominant model CT+TT 267(32.80) 285(33.29) 0.98(0.80-1.20) 1.03(0.81-1.31)

 CC 547(67.20) 571(66.71) 1.00(ref.) 1.00(ref.)

 recessive model TT 37(4.55) 27(3.15) 1.46(0.88-2.42) 1.32(0.72-2.41)

 CC+CT 777(95.45) 829(96.85) 1.00(ref.) 1.00(ref.)

rs16860234 codominant model AA 522(63.74) 573(66.63) 1.00(ref.) 1.00(ref.)

 AC 267(32.60) 254(29.53) 1.15(0.94-1.42) 1.19(0.93-1.52)

 CC 30(3.66) 33(3.84) 0.99(0.60-1.66) 0.98(0.52-1.85)

 dominant model AC+CC 297(36.26) 287(33.37) 1.14(0.93-1.39) 1.17(0.92-1.48)

 AA 522(63.74) 573(66.63) 1.00(ref.) 1.00(ref.)

 recessive model CC 30(3.66) 33(3.84) 0.95(0.58-1.58) 0.93(0.49-1.74)

 AA+AC 789(96.34) 827(96.16) 1.00(ref.) 1.00(ref.)

rs7651090 codominant model AA 445(54.00) 486(56.58) 1.00(ref.) 1.00(ref.)

 AG 321(38.96) 317(36.90) 1.11(0.90-1.35) 1.17(0.92-1.48)

 GG 58(7.04) 56(6.52) 1.13(0.77-1.67) 1.09(0.68-1.74)

 dominant model AG+GG 379(46.00) 373(43.42) 1.11(0.92-1.35) 1.16(0.92-1.45)

 AA 445(54.00) 486(56.58) 1.00(ref.) 1.00(ref.)

 recessive model GG 58(7.04) 56(6.52) 1.09(0.74-1.59) 1.02(0.65-1.62)

 AA+AG 766(92.96) 803(93.48) 1.00(ref.) 1.00(ref.)

Notes: *Logistic regression analyses adjusted for age, pre-gestational BMI and family history of diabetes; Ref, reference genotype.

5.The authors stated that "the distributions of rs1374910 in the non-GDM group deviated from HWE. How do you justify that? Because such deviation essentially reflects improper selection of controls.

Response: Thank the reviewer very much for this suggestion. Chi-square test of goodness of fit was used to measure the coincidence between the observed number of genotypes and the HWE of all genotype frequencies at the locus. If P was above 0.05, the sample of this genotype conformed to the law of genetic equilibrium, which suggested that the sample had good population representation. Since the distributions of rs1374910 in the non-GDM group deviated from HWE, we did not include this SNP in the subsequent analyses. (Line 122)

6.How do these findings help clinicians to manage GDM? It should be discussed along with the limitations of the study.

Response: Thank the reviewer very much for this advice. Since our results were all negative, clinicians still could not identify high-risk individuals by these sites. This limitation has been added in the revised manuscript. (Line 226)

7.The conclusion should be expanded and include a "Take to home" message.

Response: Thank the reviewer very much for this advice. Some suggestions are added in the revised conclusion. (Line 233)

Reviewer #2: In this study, the authors found no association of IGF2, IGF2R, and IGF2BP2 polymorphisms with gestational diabetes mellitus in Wuhan, China. The study is very well designed and written.

 1. More details on the genotyping method are required.

Response: Thank the reviewer very much for this advice. The genotyping methods have been described in detail in the revised manuscript. (Line 109)

 2. The language of the manuscript is acceptable, however, further revision might be required.

Response: Thank the reviewer very much for this advice. The language of the revised manuscript has been polished by an English Professor.

---

## [Decision Letter · Decision Letter 1]

18 Jan 2024

The associations of IGF2, IGF2R and IGF2BP2 gene polymorphisms with gestational diabetes mellitus: a case-control study

PONE-D-23-27799R1

Dear Dr. Qiong Dai,

We’re pleased to inform you that your manuscript has been judged scientifically suitable for publication and will be formally accepted for publication once it meets all outstanding technical requirements.

Kind regards,

Ramu Adela, Ph.D

Academic Editor

PLOS ONE

Additional Editor Comments (optional): No

Reviewers' comments:

Reviewer's Responses to Questions

**Comments to the Author**

1. If the authors have adequately addressed your comments raised in a previous round of review and you feel that this manuscript is now acceptable for publication, you may indicate that here to bypass the “Comments to the Author” section, enter your conflict of interest statement in the “Confidential to Editor” section, and submit your "Accept" recommendation.

Reviewer #1: All comments have been addressed

2. Is the manuscript technically sound, and do the data support the conclusions?

Reviewer #1: Yes

3. Has the statistical analysis been performed appropriately and rigorously? 

Reviewer #1: Yes

4. Have the authors made all data underlying the findings in their manuscript fully available?

Reviewer #1: Yes

5. Is the manuscript presented in an intelligible fashion and written in standard English?

Reviewer #1: Yes

6. Review Comments to the Author

Reviewer #1: The authors have satisfactorily addressed all queries and concerns raised by me. They have also appropriately revised the manuscript in light of the suggestions and concerns raised. I recommend the acceptance of the manuscript in its current form.

7. PLOS authors have the option to publish the peer review history of their article (what does this mean?). If published, this will include your full peer review and any attached files.

Reviewer #1: No

---

## [Editor Report · Acceptance letter]

25 Apr 2024

PONE-D-23-27799R1 

PLOS ONE

Dear Dr. Dai, 

I'm pleased to inform you that your manuscript has been deemed suitable for publication in PLOS ONE. Congratulations! Your manuscript is now being handed over to our production team.

Kind regards, 

on behalf of

Dr. Ramu Adela 

Academic Editor

PLOS ONE